# Single-shot polarimetry of vector beams by supervised learning

**Davide Pierangeli** [1,2] ✉ **& Claudio Conti** [2,3]

States of light encoding multiple polarizations - vector beams - offer unique capabilities in metrology and communication. However, their practical application is limited by the lack of methods for measuring many polarizations in a scalable and compact way. Here we demonstrate polarimetry of vector beams in a single shot without any polarization optics. We map the beam polarization content into a spatial intensity distribution through light scattering and exploit supervised learning for single-shot measurements of multiple polarizations. We characterize structured light encoding up to nine polarizations with accuracy beyond 95% on each Stokes parameter. The method also allows us to classify beams with an unknown number of polarization modes, a functionality missing in conventional techniques. Our findings enable a fast and compact polarimeter for polarization-structured light, a general tool that may radically impact optical devices for sensing, imaging, and computing.

Generating, manipulating, and detecting the optical state of polarization (SOP) is of paramount importance in many areas, such as optical communication[1], sensing[2], microscopy[3], and quantum information and computation[4]. While progress in material growth and nano-technology are enabling advances in active polarization control[5-7], the measurement of light polarization remains limited by its intrinsic vectorial nature. Complete determination of a single SOP needs at least four individual measurements, each projecting the state on a distinct vector[8-10]. Conventional polarimetry methods replicate in time or space the polarization analyzer, which results in bulky optical setups, or in costly compact polarimeters based on metasurfaces[11-18]. While for uniformly-polarized light the need for several measurements is still affordable and can be mitigated by using specific optical devices[19], it becomes a serious issue for beams with a spatial polarization structure.

Light with non-uniform polarization across the transverse plane exhibits non-separable correlations between polarization and spatial modes[20]. These vector beams recently have disclosed unique potentials in metrology[21-23], communication[24,25], optoelectronics[26], optomechanics[27], and quantum information[28,29]. However, their characterization still relies on bulky polarization optics[30-34]. Fast, accurate, and compact polarization measurement is crucial for exploiting vector beams in widespread applications.

In this article, we demonstrate compact single-shot measurements of multiple polarizations by photonic machine learning. We map the beam polarizations into a complex spatial distribution of intensity corresponding to a point in a high-dimensional feature space. Then, supervised learning extracts polarization information from the intensity data. The critical point is that we avoid projecting on a polarization basis and perform no direct operation on the polarization state. The method thus removes the need for polarization optics and engineered devices in polarization imaging. We report accurate measurements of various beams encoding multiple SOP, including the case in which the number of polarization modes is unknown and inferred by the measurements. An unexpected outcome is the experimental evidence of the double descent, a phenomenon that is attracting attention in machine learning[35]. The double descent increases the classification accuracy when the dimension of the feature space is large. This effect improves the observation fidelity of the multiple SOP and enables high-precision measurements.

## Results

### Single-shot polarimetry via light scattering and learning
Figure 1 reports our methodology. We show a representation of the SOP given by the four-component vector $|s\rangle = |S_1, S_2, S_3, S_0\rangle$, with Stokes parameters $S_i$. The system phase space is the unit sphere,

[1]Institute for Complex Systems - National Research Council (ISC-CNR), 00185 Rome, Italy. [2]Physics Department, Sapienza University of Rome, 00185 Rome, Italy. [3]Research Center Enrico Fermi (CREF), 00184 Rome, Italy. ✉e-mail: davide.pierangeli@roma1.infn.it

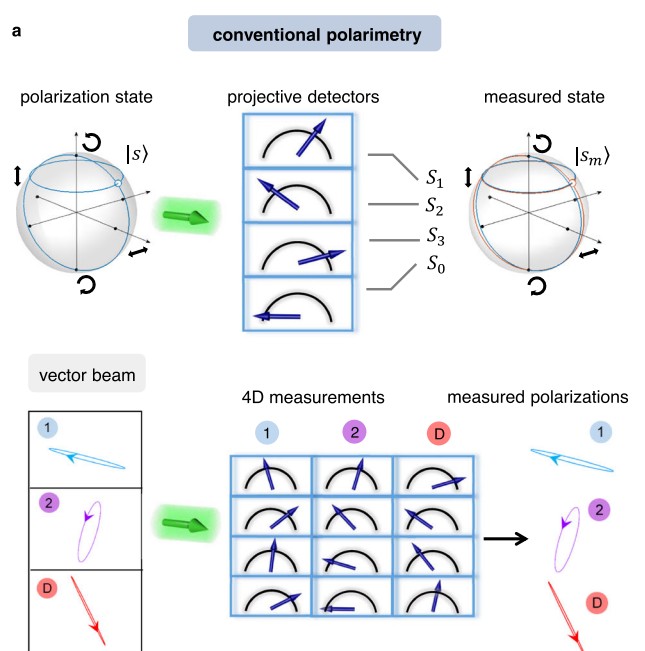

**Fig. 1 | Observing many polarizations in a single shot. a** Conventional polarimetry relies on projective measurements: the state of polarization (SOP) $|s\rangle = |S_1, S_2, S_3, S_0\rangle$, a point on the Poincaré-Bloch sphere, is processed by distinct detectors, each analyzing one component. **b** In our single-shot method, the SOP is mapped onto a high-dimensional phase space (false-color map) where data were collected. The black box represents the transformation setup, i.e., any optical system with a polarization-dependent response: disordered media, nonlinear materials, optical devices, etc. The SOP is identified in the feature space by supervised learning. When measuring a vector beam composed by 1, 2, ..., $D$ different polarizations, the projective analysis necessitates $4D$ measurements, while we use a single intensity distribution embedding all information.

known as the Poincaré-Bloch (PB) sphere. A projective measurement returns the components of the incident state $|s\rangle$ in a set of analysis vectors (Fig. 1a). Given a minimal set of four independent detected optical powers $|p\rangle = |p_1, p_2, p_3, p_4\rangle$, the measured state is obtained as $|s_m\rangle = \hat{P}^{-1}|p\rangle$, where $\hat{P}$ is the instrument matrix[9,10].

In our single-shot polarimetry (Fig. 1b), the light beam interacts with a physical object that transforms the input polarization into a set of features. The object can be any polarization-dependent optical system, such as a diffractive element, a disordered medium, etc., and its details can be unknown to the observer. The feature set at the readout is composed of an ensemble of different observable quantities. Specifically, we consider the field intensity at different $n$ spatial positions. An intensity measurement collects the values $|x\rangle = |x_1, x_2, ..., x_n\rangle$. The polarization state is determined by a linear operation as

$$|s_o\rangle = \hat{\beta}|x\rangle, \qquad (1)$$

where the $\hat{\beta}$ operator is a $4 \times n$ matrix, which we refer to as the calibration matrix of the instrument. Crucially, $\hat{\beta}$ is not determined a-priori, but retrieved by experimental data via machine learning.

The key point of our single-shot method is the redundant mapping of a polarization state, determined by four observables, to a state belonging to a much larger space, defined by $n$ observables. If $n$ is very large, the redundancy makes the method very advantageous to measure light beams that encode many polarizations in distinct spatial optical modes (vector beams). We consider a beam containing $D$ different polarization states. The corresponding multiple SOP can be expressed as $|\mathbf{s}\rangle = |s^1\rangle \oplus |s^2\rangle \oplus ... \oplus |s^D\rangle$, where $|s^j\rangle$ denotes the Stokes vector of the $j$-th mode. The observation of multiple SOP with conventional polarization tomography requires at least $4D$ projections or $4D$ generalized measurements[34]. On the contrary, we observe the state $|\mathbf{s}_o\rangle$, composed of $D$ individual SOP, with a single-shot measurement by using the higher-dimensional data vector $|x\rangle$. The

necessary calibration matrix $\hat{\beta}$ of size $4D \times n$ is determined by an initial training phase. The scheme is scalable with the dimension $D$ of the multiple SOP, at variance with projective measurement setups where additional detectors are required. In our case, we can observe input states $|s\rangle$ of variable dimension by adopting a unique feature space. This property means we can obtain, through the same detected signal, additional beam parameters otherwise difficult to access.

Single-shot polarimetry is experimentally implemented by using the scheme illustrated in Fig. 2. The optical setup is composed of two parts: a generator that produces vector beams and a single-shot polarization analyzer that realizes the optical transformation and collects the resulting intensity (details in Methods). To generate multiple SOP we exploit a phase-only spatial light modulator (SLM). This allows shaping a large number of different polarizations on the wavefront of a single $\lambda = 532$ nm laser beam, with distinct states $|s^j\rangle$ that correspond to spatially-separated modes with phase $\phi_j$ (see Supplementary Note 1). The single-shot analyzer exploits light scattering[36]. In fact, it is known that multiple scattering from random media can be harnessed to perform deterministic operations on an incident electromagnetic field[37]. The idea have resulted in optical instruments based on disorder, such as lenses[38], polarimeters[39], beam analyzers[40], and compact spectrometers[41], but also in effective schemes for optical computing[42]. In our experiment, we use light scattering from a glass diffuser to map the SOP into intensity data. The scattering medium spatially mixes the impinging optical field and transmits a disordered field. The resulting speckle pattern is imaged on a camera sensor with no polarization filters. The entire polarization content of the incoming beam is hidden within this intensity distribution. However, at variance from disorder-based photonic instruments[39–41], our method does not require determining the transmission operator to perform the measurement. We simply get an image whose spatial details depend on the polarizations of the vector beam. By sampling the image, we obtain the output data $|x\rangle$, which encodes the input polarizations into a higher-dimensional feature space. The mapping can be performed by any optical system

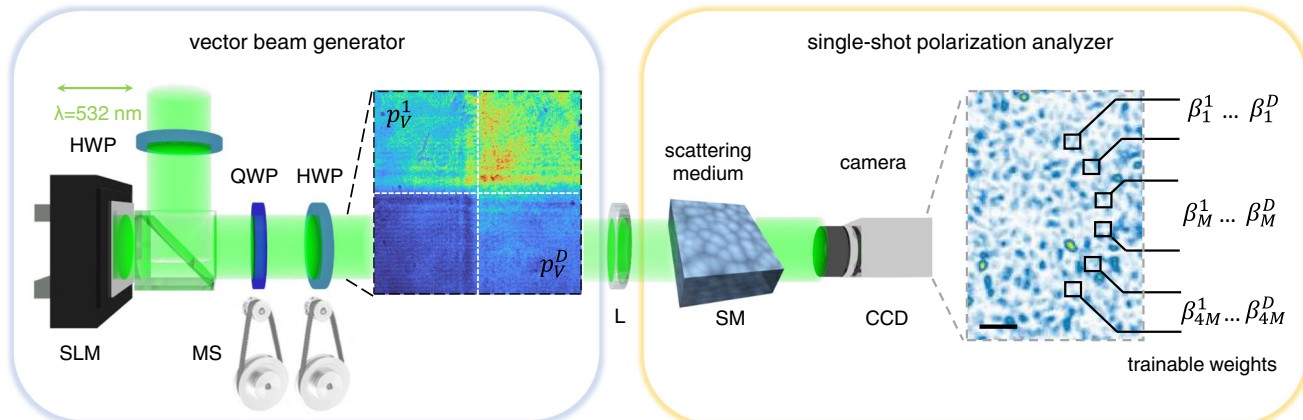

**Fig. 2 | Experimental setup.** The vector beam generator is based on a phase-only spatial light modulator (SLM). Inset shows the vertical intensity component $p_V$ of a partitioned beam composed of four SOP. The apparatus for single-shot polarimetry is composed of a scattering medium (SM) and a camera (CCD). The collected spatial intensity distribution (false-color inset) is processed on $4M$ linear output channels that enable training via the readout weights $\beta_i^j$. HWP half-plate, QWP quarter-waveplate, LP linear polarizer, MS motor stages, L lens.

sensitive to the input polarization such as optical fibers and nonlinear materials.

To understand the role of coupling between polarization and spatial degrees of freedom, we differentiate two cases. For $D = 1$, a necessary working condition is a scatterer that couples polarization and amplitude spatial modes. In fact, different input SOP have to result in distinguishable intensity distributions to obtain distinct measurement outcomes. Surprisingly, for vector beams ($D > 1$), coupling generated by the scatterer is no longer necessary. In this case, the partitioning of the impinging beam induces sufficient interaction (see Supplementary Note 2). In experiments, we vary the polarization-spatial coupling by testing glass diffusers of various roughness, which induce a different amount of depolarization[43]. We found comparable performance in all the considered samples. Hereafter, we report results in the case of small coupling, corresponding to transmitted light with a degree of polarization close to one (see Supplementary Note 3).

To extract polarization information from the intensity data, we use supervised learning. Among the various neural network architectures that have been proved convenient for optical systems[44], we adopt the extreme learning machine (ELM)[45]. ELM allows fast and easy training with several thousands of network nodes, thus being especially suited to large-scale photonic implementations[46–54]. To apply the ELM algorithm, we construct a readout layer by a random selection of $4M$ output camera channels (Fig. 2b). Each channel has a linear weight $\beta_i^j$, with $i = 1, ..., 4M$ and $j = 1, ..., D$, which form the calibration matrix in Eq. (1). Training consists in adjusting these parameters according to a labeled dataset via ridge regression (see Methods).

## Machine-learning polarimeter

Figure 3 shows the single-shot polarimetry of a single SOP. We generate $N_{train}$ samples randomly distributed on the PB sphere (Fig. 3a) and validate the calibrated single-shot analyzer on $N_{test}$ unknown SOP. As reported in Fig. 3b, the observed polarizations correspond to the input within a distance $d = 0.014 \pm 0.002$ (see Methods). Each measured Stokes parameter $S_i$ is in remarkable agreement with its target value (Fig. 3c). The accuracy of the machine-learning polarimeter is investigated by varying the number of network channels $M$ and the size $N_{train}$ of the calibration dataset. Results in Fig. 3d show the behavior of the Stokes error $E(S_i)$. We find that single-shot measurements become extremely accurate when $M$ increases ($E(S_i) = 0.0075$).

The observed error peak in Fig. 3d discloses the so-called double descent[35]. Above the interpolation threshold, the accuracy increases with the number of channels with no overfitting. We find that the interpolation threshold shifts with the training dataset size (Fig. 3d), with a maximum error for $M \simeq N_{train}$. This point marks a critical condition in ELM made with physical systems[48]. Experimental observation of the double descent is made possible by the huge number of output nodes of our optical setup[54]. We exploit the effect to enhance the polarimeter precision. On the other hand, we underline that a few hundred channels are sufficient for effective measurements. Figure 3f shows the accuracy matrix $\hat{a}$, which also includes the degree of polarization $v$. The single-shot results match well data from a conventional rotating-waveplate polarimeter. With reference to single-qubit tomography[55], we obtain a fidelity $F(\rho_s, \rho_m) = 0.99 \pm 0.01$, where $\rho_s$ and $\rho_m$ are the density matrices obtained via single-shot and multiple projections.

## Single-shot polarimetry of vector beams

In Fig. 4, we report single-shot polarization measurements of partitioned vector beams with four SOP. Calibration operates alike the single polarization case, but with $N_{train}$ states of dimension $D = 4$. Figure 4a shows the intensity of an unknown multiple SOP generated by four random phases $\phi_i$ on four spatial modes. The effect of diffraction between adjacent SOP is visible. By collecting the transformed intensity image, we reveal the full polarization structure in a single shot, as shown by the polarization ellipses in Fig. 4b. Conventional projective polarization tomography (Fig. 4c) strikingly agrees with the single-shot observation. The overall uncertainty of the detection for $D = 4$ is evaluated in Fig. 4d by varying $M$ up to $4M = 52,000$ total channels. The error decreases with $M$, with localized peaks that indicate the interpolation threshold. The fidelity is confirmed by the overlap matrix $\hat{a}$ in Fig. 4e, with $\mathrm{Tr}(\hat{a})/4D = 0.95$ accuracy.

We succeed in characterizing with high accuracy beams encoding up to nine SOP. This implies a single-shot reconstruction of a classical state $|s\rangle$ lying in a 27-dimensional phase space. To implement larger systems, we expand the spatial extent of the input and output optical planes. Figure 5a shows the intensity of a beam encoding multiple SOP with $D = 9$. The result of the single-shot detection agrees with the state we get from multiple projections (Fig. 5b, c). The comparison, with the $\hat{a}$ matrix having 1296 entries, gives a 0.91 overlap between the polarization states measured by using the two methods. The single-shot observation is hence performed with high precision, as also the measurement error in Fig. 5d indicates. Interestingly, we observe that the distance $d$ decreases rapidly with $M$ and gets stuck into a broad plateau. This behavior underlines the enormous complexity of the states that we are characterizing. Even if, in Fig. 5d, we do not observe any

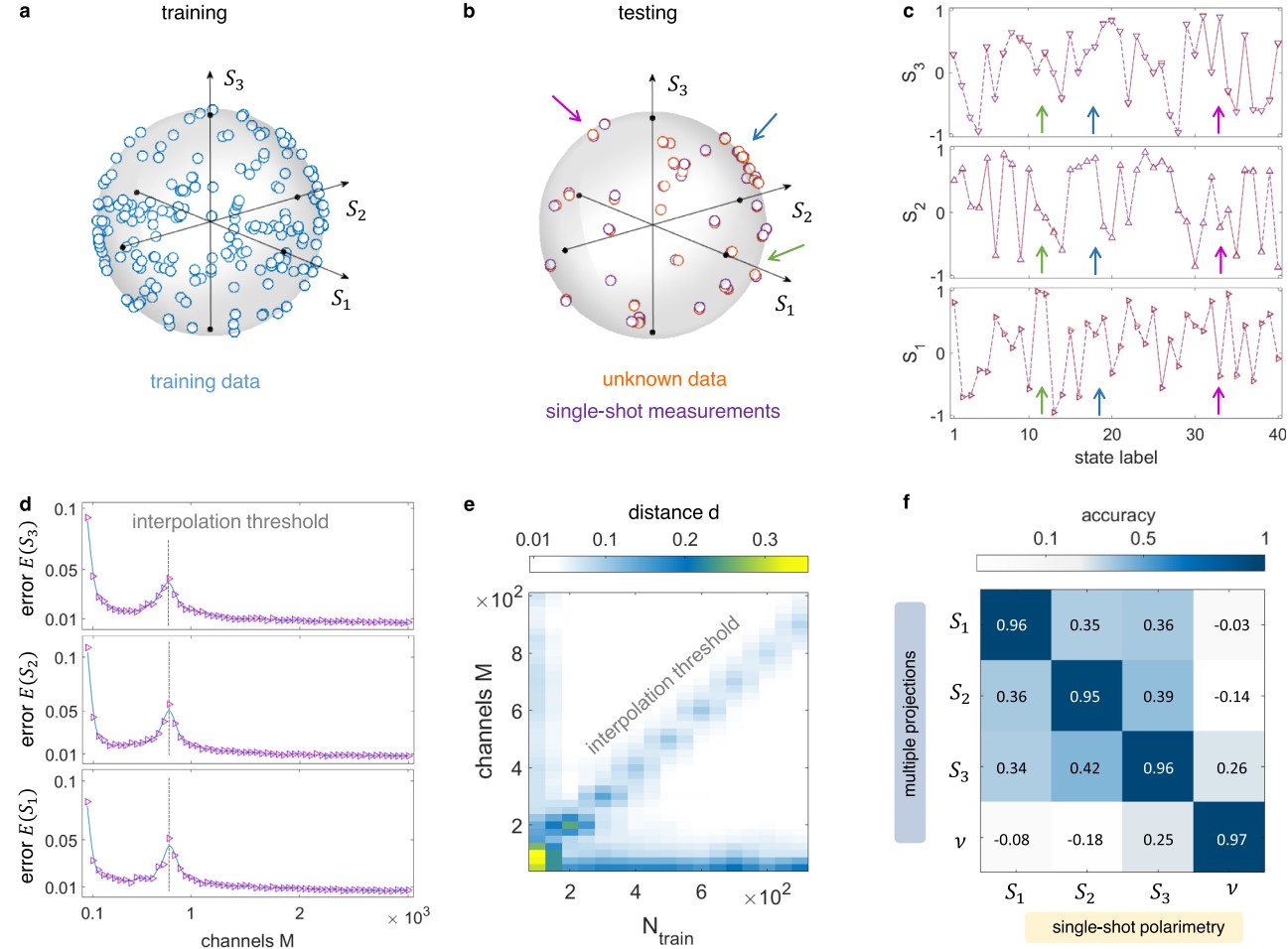

**Fig. 3 | Polarization measurements by supervised learning. a** Training dataset made by $N_{train}$ randomly-selected polarizations. **b**, **c** Measured SOP and corresponding Stokes parameters (purple), along with their unknown values used for testing (orange). For clarity, only a subset of samples is shown. **d** Testing error $E(S_i)$ on each Stokes parameter versus the number of readout channels. The error peak (interpolation threshold) is direct evidence of the double-descent effect. **e** Distance between the measured and generated SOP varying the number of training samples and channels. **f** Accuracy matrix (see Methods) comparing single-shot ($M = 200$) and conventional polarimetry performed via multiple projections.

interpolation threshold, as more output channels would be necessary, the method still enables accurate measurements.

## Measuring the number of polarizations

In the most general case of an unknown multiple-polarization state, its dimension $D$ is also an unknown variable. We demonstrate that we can identify the dimensionality of the multiple SOP. To train the setup, the calibration dataset includes beams with a variable number of partitions, and $D$ as an additional target parameter. Figure 6a reports the confusion matrix obtained for vector beams having three possible partition configurations ($D = 1$, $D = 4$, and $D = 9$), with $M = 10,000$. The dimension is found with an accuracy that exceeds 98%. In addition, the set of Stokes parameters is measured with precision comparable to cases of known dimension ($E(S_i) = 0.057$). For $D = 9$, a comparison between the entire $S_i$ distribution is shown in Fig. 6b. Therefore, we not only perform single-shot polarimetry of beams encoding nine polarizations, but we carry out the measurement not knowing that the overall polarization vector has nine dimensions. It is important to note that dimension $D$ is not directly measurable by using multiple projective measurements.

## Discussion

We have experimentally demonstrated the measurement of multiple polarizations in a single shot without polarization optics. The result has been obtained with an original method that combines physical transformation between polarization and spatial degrees of freedom with machine learning to get unprecedented information in a single detection. This approach demonstrates photonic machine learning for a challenging optical application, i.e., polarization imaging.

Our measurement scheme has no bulky optical components and can operate at any wavelength. This allows to overcome the narrowband operation of all polarization cameras based on integrated devices and metasurfaces[17,18]. In our setup, both the scattering process and the learning procedure are efficient on a broad spectrum. As the transmitted intensity distribution is highly sensitive to the incoming spectrum, we expect our polarization imaging is able to operate on broadband light by training a wavelength-dependent instrument matrix $\hat{\beta}(\lambda)$. The single-shot polarimeter is hence compact, without moving components, broadband, and does not require nanofabrication. Moreover, it provides direct access to properties of the vector field otherwise difficult to quantify, as we demonstrate by measuring the unknown number of polarizations within the vector beam. The device can be developed further by using unsupervised learning methods such as variational autoencoders that generate Stokes parameter distributions from the detected sparse data.

These findings empower compact single-shot polarimetry based on machine learning in a wide variety of contexts, from optical networking to biomedical devices. We foresee the extension of our

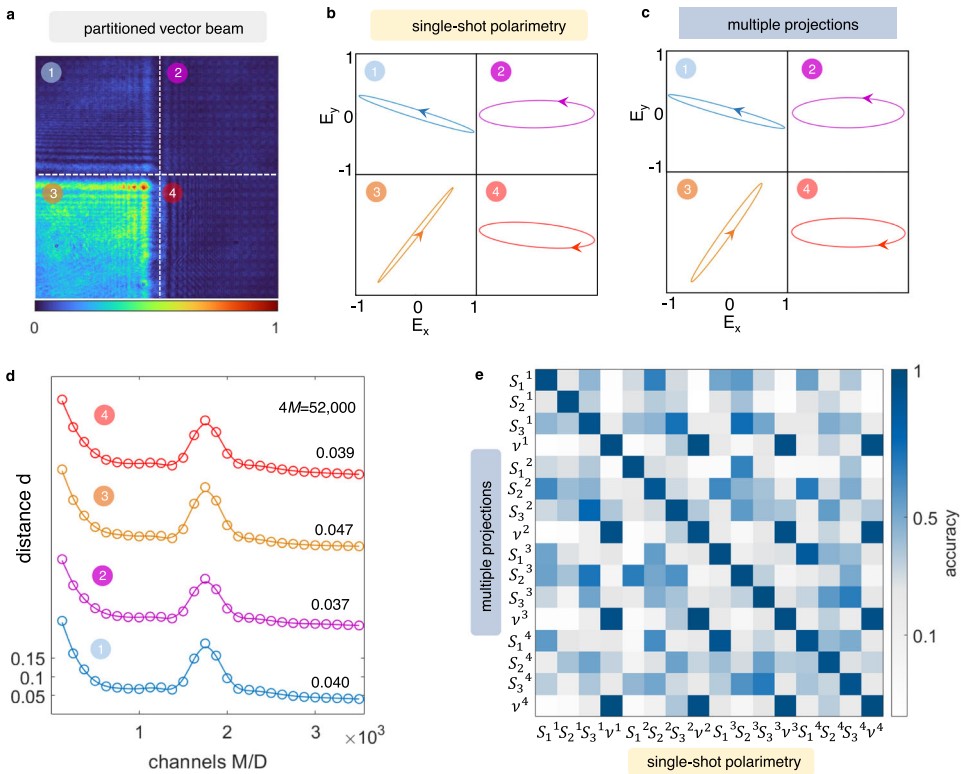

**Fig. 4 | Single-shot polarimetry of four-parted vector beams. a** A generated beam with $D = 4$. We show the spatially-resolved vertical intensity component $p_V$. **b** Polarization ellipses observed in a single shot ($M/D = 1000$) and **c** via multiple projective measurements. **d** Distance of the $j$-th mode as a function of the number of channels $M$. Data have been shifted vertically for clarity; inset values are the minimum $d$ for each curve. **e** Accuracy matrix of the single-shot polarimetry for $D = 4$.

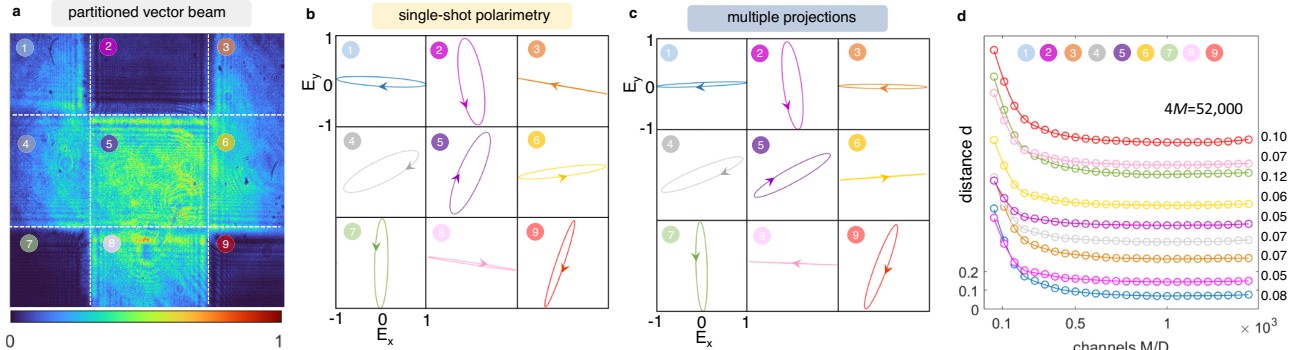

**Fig. 5 | Single-shot polarimetry of nine-parted vector beams. a** Intensity $p_V$ measured when vertically projecting a generated beam with $D = 9$. The total state $|\mathbf{s}\rangle$ belongs to a 27-dimensional phase space. **b** Single-shot measurement of the nine polarizations. **c** Polarization tomography by multiple measurements, for comparison. **d** Distance between the single-shot observations and generated SOP. Curves are shifted for visualization; the inset shows their value for 52,000 channels.

single-shot approach to the entire electromagnetic spectrum[56], to subwavelength and topological optical fields, and, more generally, to other optical degrees of freedom[57], with applications where conventional instruments are useless, as in edge devices and photonic chips[58]. Moreover, partitioned vector beams of quantum light can encode many qubits. Therefore, our results may also open exciting perspectives in the quantum domain, with the possibility to benefit from single-shot polarimetry by machine learning.

## Methods
### Experimental setup

The experimental setup follows the sketch in Fig. 2. A continuous-wave laser beam with wavelength $\lambda = 532$ nm (LaserQuantum Ventus 532, 250 mW) is expanded and linearly polarized along the horizontal (H) direction ($x$-axis) with a linear polarizer (LP). The vector beam generator is composed of a reflective phase-only SLM (Hamamatsu X13138, $1280 \times 1024$ pixels, 12.5 μm pixel pitch, 60Hz frame rate) sandwiched between an input half-waveplate (HWP) and an output polarization system made by a quarter-waveplate (QWP) and HWP. The output waveplates are equipped with motorized precision rotation stages (MS) (25°/s maximum rotation velocity) that are programmed online. Their fast axes form respectively an angle $\alpha$ and $\beta$ with the $x$-axis. The parameters $\alpha$ and $\beta$ are varied with 2° resolutions during both training and testing. By grouping $L \times L$ SLM pixels, the modulator active area is divided into $D$ squared input modes, with each mode having a phase $\phi_j$ in the [0, 2π]

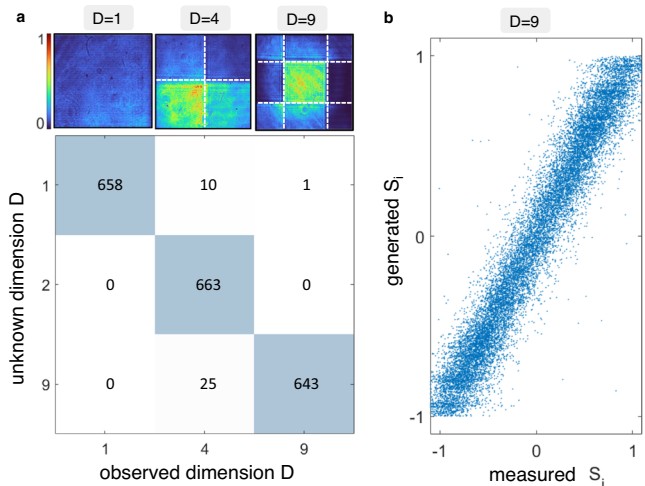

**Fig. 6 | Measurement of a beam of unknown polarization number. a** Confusion matrix over $N_{test} = 2000$ beams with three possible values for $D$. Insets are representative projective images in the three cases. $D$ is determined with 98% accuracy. **b** Single-shot measured versus generated Stokes parameters for beams of nine parts, whose dimension has been automatically identified.

interval. The available phase levels are 210, distributed according to a linear phase response curve[59]. Polarization-modulated light is focused by a plano-convex lens (f = 150 mm) on a scattering medium (Thorlabs N-BK7 Ground Glass Diffusers, 1500 grit) positioned using a four-axis translational stage. The scattered field is collected by an imaging objective (NA = 0.25) and the transmitted speckle pattern is detected by a cooled camera (Basler a2A1920-160umPRO, 1920 × 1200 pixels, 160 fps) with 12-bit (4096 gray-levels) intensity sensitivity. Within the camera region of interest, $4M$ output channels are randomly pre-selected. The signal is obtained by binning over a few camera pixels (4 × 4 pixels) to reduce detection noise. Output channels have a size comparable to the spatial extent of a speckle grain. For reference measurements, a portion of the polarization-modulated beam is split and analyzed by using conventional polarimetry. A commercial rotating-waveplate polarimeter (PAX1000VIS, 0.25° accuracy) is used for a single SOP. For projective polarization imaging of vector beams, we use a custom rotating-waveplate analyzer composed of a QWP, an LP, and a CMOS camera (DCC1545M). Intensity projections are measured along the horizontal (H), vertical (V), diagonal (D), and right circular (R) components, i.e., $p_H$, $p_V$, $p_D$, and $p_R$ are the detected power. An example of projective analysis for a partitioned vector beam is shown in Supplementary Fig. 2.

## Training method
Calibration of the setup for single-shot polarization measurements is performed by generating $N_{train}$ vector beams and by loading one-by-one the corresponding phase mask on the SLM. For the $l$-th training sample, the output waveplates of the vector beam generator are rotated to a couple of parameters $\alpha_l$ and $\beta_l$ (Supplementary Note 1). The dataset $\{\phi_l^j, \alpha_l, \beta_l\}$ is randomly generated to cover the entire phase space. When the polarization dimension $D$ need also to be determined, the training set is composed of states whose number of polarizations varies within the dataset. Intensity values from the $4M$ output channels are stored (Supplementary Fig. 6). We increase linearly the number of selected channels when increasing the dimension. In the case of $D = 9$, for the feasibility of training, we consider a maximum value of $M = 13,000$. For the measurement of multiple SOP with an unknown polarization number, $M$ additional channels are used to reconstruct $D$. The calibration weights $\beta_i^j$ are determined by applying the training algorithm to the entire set of acquired data. The obtained

calibration matrix $\hat{\beta}$ is used to measure $N_{test}$ random SOP. We use a ratio $N_{test}/N_{train} = 0.1$.

## Extracting polarizations from intensity data
In our setup, the scattering medium creates a mapping between the incoming polarization set and a higher-dimensional feature space. This general mechanism underlies physical implementations of ELM and kernel machines for neuromorphic computing[47,49,52]. In our case, the scheme is trained to perform physical measurements. To determine the calibration matrix $\hat{\beta}$ by using $4M$ output channels, we consider a training set of randomly-selected Stokes vectors $\{|s\rangle\} \equiv \mathbf{S}$ as target vectors ($N_{train} \times 4D$ sized). The corresponding acquired intensity matrix is $\{|x\rangle\} \equiv \mathbf{X}$, with size $N_{train} \times 4M$ ($M \gg D$). Training corresponds to solving numerically the ridge regression problem:

$$\arg\min_{\hat{\beta}}(\| \mathbf{X}\hat{\beta} - \mathbf{S}\|^2 + c^{-1} \| \hat{\beta}\|^2), \tag{2}$$

where parameter $c$ controls the trade-off between the training error and the regularization. A solution is given by[45]

$$\hat{\beta} = \left(\mathbf{X}^T\mathbf{X} + c\mathbf{I}\right)^{-1}\mathbf{X}^T\mathbf{S}, \tag{3}$$

where $\mathbf{I}$ is the identity matrix. Inversion involves the $4M \times 4M$ matrix $\mathbf{X}^T\mathbf{X}$, which makes the method scalable as $M$ is selected by the observer. Therefore, given an unknown multiple SOP and the corresponding single-shot intensity vector $|x\rangle$, the observed state can be expressed as

$$|\mathbf{s}_o\rangle = \left(\mathbf{X}^T\mathbf{X} + c\mathbf{I}\right)^{-1}\mathbf{X}^T\mathbf{S}|x\rangle. \tag{4}$$

An explicit expression for $x_i$ is reported in Supplementary Note 2. In the case of a single SOP, the measured Stokes parameters from single-shot intensity data are

$$S_i = \sum_{k=1}^{M} \beta_k x_k. \tag{5}$$

Each $S_i$ is hence decoded through $M$ weights. An example of both the acquired intensity and calibration matrix ($c = 1$) is reported in Supplementary Note 4.

## Analysis of the measurements
To quantify the accuracy of the single-shot polarimeter, as a testing error on the $i$-th Stokes parameter, we use the mean-absolute-error (MAE) $E(S_i) = \langle |S_i^s - S_i^g| \rangle$, in which the apex stands for single-shot (s) and generated (g), and the average is over $N_{test}$ samples. Instead of $S_0$, we report the degree of polarization $\nu = \sqrt{S_1^2 + S_2^2 + S_3^2}/S_0$, which quantifies the amount of unpolarized light. The distance between two SOP on the PB sphere is computed as $d = \sqrt{\sum_i E(S_i)^2}$. The accuracy values with respect to projective measurements are $a_{ij} = 1 - O_{ij}$, with the overlap $O_{ij} = \langle |S_i^s - S_i^m| \rangle$, with apex $m$ stands for multiple measurements. It is worth noting that this comparison includes also the inaccuracy of the SOP generator and the uncertainty of the conventional polarization analyzer (Supplementary Fig. 2). For beams with a single SOP, the fidelity is computed in analogy with quantum state tomography as[55] $F(\rho_s, \rho_m) = \text{Tr}\left(\sqrt{\sqrt{\rho_m}\rho_s\sqrt{\rho_m}}\right)^2$, where the density matrix is $\rho = 1/2(\sum_i S_i \hat{\sigma}_i)$ with Pauli matrices $\hat{\sigma}_i$.

## Reporting summary
Further information on research design is available in the Nature Portfolio Reporting Summary linked to this article.

## Data availability

All relevant data that support the findings of this study are available within the article and supplementary information. Any additional data are available from the corresponding author upon request.

## Code availability

The code used in the present study are available from the corresponding author upon request.

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

## Acknowledgements

We acknowledge funding from the Italian Ministry of University and Research (PRIN PELM 20177PSCKT, C.C.), and FLAG-ERA JTC 2019 (MARGO). We thank L. Dieli for useful discussions, I. MD Deen and F. Farrelly for technical support in the laboratory.

## Author contributions

D.P. and C.C. developed the idea and wrote the manuscript. D.P. carried out the experiments and data analysis.

## Competing interests

The authors declare no competing interests.
