## [Peer Review File · Nature Communications]

Single-shot polarimetry of vector beams by supervised learningReviewer #1 (Remarks to the Author):

The authors describe a projective measurement that returns the components of the incident state of polarization of an incoming electromagnetic field.

It is claimed that (1) a scattering medium can be used to map the beam polarizations into a complex spatial distribution of intensity and (2) that, because of this, the technique does not require specific polarization optics elements.

The authors present a careful description of the process and some experimental results.

The main problem is the novelty. The fact that the transmission matrix of a scattering medium can be effectively calibrated for and subsequently used for performing deterministic operations on an incident electromagnetic field has been known for a long time starting from the ideas put forward in *Physica A* 168, 49, 1990. There is an abundant literature regarding this topic, a few notable works are in *OL* 32, 2309, 2007 and *PRL* 104, 100601, 2010 among many others, ending up with a very recent comprehensive review in *Nature Physics* 18, 2022.

In the past, upon systematic calibration or training to infer the transfer characteristics, disordered scattering systems have been used as spectrometers (see for instance *Nature Photonics* 7, 746, 2013). In fact, random scattering and appropriate training algorithms have been used to actually measure exactly the polarimetric characteristics even before that (*Opt Express* 16, 13232, 2008). What's more, using similar approaches it was demonstrated the possibility to determine even spatially resolved polarimetric information, *OL* 34, 1321, 2009, or to infer, simultaneously, both the spectral and polarimetric content of an incoming field.

In view of all the significant and closely related body of prior work, the impact of the present manuscript is rather limited. Therefore, I cannot recommend publication in *Nature Communications*.

Reviewer #2 (Remarks to the Author):

Please, see uploaded file

Reviewer #3 (Remarks to the Author):

What are the noteworthy results?

This manuscript proposes an innovative and clever idea for measuring the point-to-point polarization state of an unknown vector beam at the input. The idea is to create a sparse representation of the data by light scattering in a random medium and then use machine learning to extract features. These features predict the resulting polarization state. The authors perform a rigorous theory and verify the system against experiments, reporting prediction results with accuracies beyond 95%, representing a notable outcome.

Will the work be of significance to the field and related fields? How does it compare to the established literature? If the work is not original, please provide relevant references.

The authors are well aware of the state-of-the-art in this field, which the group of Federico Capasso primarily advances with metasurfaces. The authors provided a deep experimental comparison against established and available techniques in Fig. 3-4. The proposed method is very competitive in terms of prediction accuracy. It retains all the advantages of using universal approximators, such as e.g., scalability, integrability, and,

more importantly, speed, and as such it reports a quite considerable improvement.

Does the work support the conclusions and claims, or is additional evidence needed?

The work is well-written, and the narrative is straightforward. One question I have is how broadband this system is. One of the limitations of nearly all the polarization cameras proposed with integrated concepts such as metasurfaces lies in their narrowband operating mode. Because of this system's universal nature, which essentially processes data and uses broadband optics, I expect it to work beyond single-frequency conditions. Can the authors comment on this point? Even a ballpark discussion will be enough if additional experiments are not possible.

Are there any flaws in the data analysis, interpretation and conclusions? Do these prohibit publication or require revision?

Is the methodology sound? Does the work meet the expected standards in your field?

The analysis is sound, and the conclusion is clear. The work definitely merits publication in NCOMM.

Is there enough detail provided in the methods for the work to be reproduced?

This part can be improved. Fig. 1 mixes different systems and is hard to grasp at first (I had to see it a few times). I would suggest the authors restructure the panels, having projective measurements in a brief part and then the proposed method in another part, with no mixing. Also, there are few details in panel b) on how the sparse representation of the data, which is the key idea here, is generated. I suggest to provide more details in this part.

Fig. 2: there are three different setups, but it is not straightforward how they are connected. Can the authors simplify here? Because it is a single-shot imaging system, so I expect a single setup?

Additional remarks:

1) I find the reported prediction accuracies very noteworthy results. I suggest that the authors comment on those in the abstract to improve the work's quantitative analysis.

2) In the ridge regression problem, how do the authors estimate the condition number c ?

3) From my understanding, the authors directly train the data's sparse representation by linear regression. I think a possibly exciting way to improve the prediction capability of this system is to use statistical inference on the sparse data generated by the scattering medium. This step could be performed, e.g., with a probabilistic autoencoder. Is this a possibility? Can the authors comment on this point?

Response to Reviewer #1

Report #1: The authors describe a projective measurement that returns the components of the incident state of polarization of an incoming electromagnetic field. It is claimed that (1) a scattering medium can be used to map the beam polarizations into a complex spatial distribution of intensity and (2) that, because of this, the technique does not require specific polarization optics elements. The authors present a careful description of the process and some experimental results.

Response: We thank the Reviewer for the time they dedicated to reviewing our manuscript. We understand from their report that important points of our work were not sufficiently clear and we here explain them in detail. To be clear, our main claim is a scheme to measure a non-uniform polarization distribution in a single shot without polarization optics. A similar method is not currently available, as explained below.

Report #1: The main problem is the novelty. The fact that the transmission matrix of a scattering medium can be effectively calibrated for and subsequently used for performing deterministic operations on an incident electromagnetic field has been known for a long time starting from the ideas put forward in Physica A 168, 49, 1990. There is an abundant literature regarding this topic, a few notable works are in OL 32, 2309, 2007 and PRL 104, 100601, 2010 among many others, ending up with a very recent comprehensive review in Nature Physics 18, 2022. In the past, upon systematic calibration or training to infer the transfer characteristics, disordered scattering systems have been used as spectrometers (see for instance Nature Photonics 7, 746, 2013).

Response: We know the many applications enabled by the measurement and control of the transmission matrix of complex media. However, in our work, we exploit scattering to achieve a completely new functionality: polarimetry of vector beams. The crucial point is that we do not make simply a measurement of the wave polarization by using disorder, but we characterize a vector beam containing many polarizations. Machine learning is the novel ingredient that is decisive to the scope. It makes our approach advantageous over all existing techniques to measure polarization-shaped light by enabling characterization (i) in a single shot (ii) with no polarization optics. Polarization imaging by machine learning was never demonstrated before. The achievement has remarkable importance as compact, fast, and broadband polarization analyzers are not currently available.

Report #1: In fact, random scattering and appropriate training algorithms have been used to actually measure exactly the polarimetric characteristics even before that (Opt Express 16, 13232, 2008). What's more, using similar approaches it was demonstrated the possibility to determine even spatially resolved polarimetric information, OL 34, 1321, 2009, or to infer, simultaneously, both the spectral and polarimetric content of an incoming field.

Response: We thank the Reviewer for pointing out these interesting articles, which have been now included in our references. (change 1) However, although these methods employ light scattering, they apply to laser beams with uniform polarization and still rely on polarization optics, in drastic contrast with our approach. To be specific, "Opt. Express 16, 13232, 2008" does not report measurements of non-uniform beams but only the recovery of a single polarization state after the complete determination of the field transmission matrix. In "Opt. Lett. 34, 1321, 2009", this approach is developed further to perform polarization imaging using a fiber polarimeter, but spatial resolution is achieved by independent scattering fibers, i.e., by replicating the analyzer in space and using multiple measurements like in conventional polarimetry. Our concept is completely different: we use a single transformation and machine learning to obtain spatially-resolved polarization information in a single shot. Moreover, in "Opt. Lett. 34, 1321, 2009" polarization optics is necessary to analyze the output field and to

characterize fiber channels. We stress that, differently from all the reported polarimetry techniques, our method does not use any polarization optics. As recognized by the other Reviewers, this crucial difference makes our single-shot polarimeter based on machine learning an unprecedented instrument that is simple, fast, and compact.

Report #1: In view of all the significant and closely related body of prior work, the impact of the present manuscript is rather limited. Therefore, I cannot recommend publication in Nature Communications.

Response: As we have discussed, no previous work reported the concepts and functionalities we demonstrate. The problem of measuring the polarization content of a vector beam efficiently is an open challenge, as certified by recent works of various groups on the topic, for example “B. Zhao et al., Opt. Express 27, 31087 (2019)”, “A. Manthalkar et al., Opt. Lett. 45, 2319 (2020)”, “M.A. Al Khafaji et al., Opt. Express 30, 22396 (2022)”, “C. He et al., Optica 9, 1109 (2022)”. In these articles, Authors clearly outline the difficulty and importance of measuring in a single shot a polarization-structured beam.

Our findings have a drastic impact on this active research, and they can find widespread application in the many different fields where fast and compact polarization measurements are vital such as in communication with structured light. Our results are especially important also for the rapidly growing field of photonic machine learning. In fact, for the first time, we apply a combination of optical mapping and digital machine learning for performing an applicative task rather than a computing task. Note that polarization imaging is otherwise unfeasible by computational methods only. The revised version of the manuscript further underlines these key aspects. (change 2)

Considering all these arguments, we hope the Reviewer can now recognize the novelty and impact of our work.

Response to Reviewer #2

Report #2: This is an interesting manuscript presenting a good mixing of theoretical and experimental results. The main idea is the following: in an ordinary polarimetry setup, the polarization state of a beam of light is assessed by measuring the so-called Stokes parameters of the beam. Such a measurement requires projecting the beam across at least four different polarizers. Vice versa, in the scheme proposed by Pierangeli and Conti, the use of polarizers and projection measurements is not required. Their technique uses a diffuser screen that creates a speckle pattern within which, somehow, is encoded (hidden, I would say) the polarization of the incoming beam. Then, with spatially-resolved intensity measurements and a suitable machine-learning algorithm, the authors are able to infer the polarization state of the input beam, even when the latter has a non-uniform polarization pattern ($D = 4$ and $D = 9$ cases).

Response: We thank the Reviewer for the profound analysis of our work. We are pleased that they completely grasped and appreciated our approach. In the revised version, we have followed their suggestions to improve the clarity of the manuscript.

Report #2: How does it work? It looks a bit of magic because to encode the polarization information within the speckle pattern, it would be necessary that the diffuser could generate a coupling between the polarization and the intensity distribution of the light. However, the authors state at page 4 that the coupling is very small. This is very surprising to me. I suspect that if this is true, then this technique could work because of the segmentation of the impinging beam when $D = 4$ and $D = 9$: this could already create a coupling between polarization and spatial degrees of freedom. It would be therefore desirable that the authors could show an

application of their technique to the case $D = 1$. If my hypothesis is correct, in this case this technique would not work well. This ambiguity about coupling/not coupling is also present in the supplemental material, where Eqs. (7) and (8) do not look consistent each other.

Response: The Reviewer is right. We have now clarified this crucial and delicate point by adding a paragraph in which we discuss in detail the role of coupling between polarization and spatial intensity modes. (change 3)

As the Reviewer noted, coupling induced by the scatterer is a necessary condition for the correct operation of our method on uniform beams. Intuitively, different SOP must result in distinguishable intensity distributions to obtain distinct measurement results. The remarkable performance of our single-shot polarimeter for $D=1$ (Figure 3) guarantees that such a coupling is present. However, we find that the coupling magnitude is not a significant parameter for the effectiveness of the method, a fact very surprising also for us. For $D=1$, the scheme works equally well (the same testing error) both when using thick diffusers (strong coupling) and in conditions of small coupling, i.e., when the variation of polarization within the speckle pattern is very small and diagonal TM elements have similar values (see Supplementary Figure 3 and Figure 4). This indicates that the machine-learning procedure can identify and classify the tiny variations of the intensity pattern resulting from the input SOP.

The above physical picture changes in the case of a vector beam ($D>1$). Surprisingly, the partitioning of the input beams makes the coupling induced by the scatterer no more necessary, as outlined by the Referee. We have corrected the model in the Supplementary Information to clarify these aspects. (change 4)

Report #2: In summary, in my opinion this manuscript deserves to be published in Nature Communications because of its quality, originality and timeliness. However, I would be glad if the authors could clarify the issues I remarked in my comment.

Response: We thank the Reviewer for the positive recommendation and valuable comments that helped us to improve the manuscript. We hope the Reviewer finds the role of coupling clarified.

Response to Reviewer #3

Report #3: This manuscript proposes an innovative and clever idea for measuring the point-to-point polarization state of an unknown vector beam at the input. The idea is to create a sparse representation of the data by light scattering in a random medium and then use machine learning to extract features. These features predict the resulting polarization state. The authors perform a rigorous theory and verify the system against experiments, reporting prediction results with accuracies beyond 95%, representing a notable outcome.

Response: We thank the Reviewer for the positive and very constructive report. We are pleased that they completely understand our results. The revised paper has been improved following their useful suggestions.

Report #3: The authors are well aware of the state-of-the-art in this field, which the group of Federico Capasso primarily advances with metasurfaces. The authors provided a deep experimental comparison against established and available techniques in Fig. 3-4. The proposed method is very competitive in terms of prediction accuracy. It retains all the advantages of using universal approximators, such as e.g., scalability, integrability, and, more importantly, speed, and as such it reports a quite considerable improvement.

Response: We thank the Reviewer for recognizing the novelty and the fact that our work is properly settled in the state-of-the-art. In response to the report of Reviewer #1, we have expanded the state-of-the-art by adding relevant references and underlining the significance of our results for various fields. (change 1) (change 2)

Report #3: The work is well-written, and the narrative is straightforward. One question I have is how broadband this system is. One of the limitations of nearly all the polarization cameras proposed with integrated concepts such as metasurfaces lies in their narrowband operating mode. Because of this system's universal nature, which essentially processes data and uses broadband optics, I expect it to work beyond single-frequency conditions. Can the authors comment on this point? Even a ballpark discussion will be enough if additional experiments are not possible.

Response: The Reviewer has centered a critical point. Our system can operate at any wavelength without major limitations. All the ingredients of the scheme – random scattering, intensity detection, machine learning on intensity data, etc. – operate in the same way at any wavelength. In future work, we aim at demonstrating our approach to broadband visible light, which is an important development as noted by the Reviewer. In this case, we expect to train a calibration tensor that also depends on the wavelength, which allows processing simultaneously many frequencies only by acting at the software level. This aspect has been discussed in the revised manuscript. (change 5)

Report #3: The analysis is sound, and the conclusion is clear. The work definitely merits publication in NCOMM.

Is there enough detail provided in the methods for the work to be reproduced?

This part can be improved. Fig. 1 mixes different systems and is hard to grasp at first (I had to see it a few times). I would suggest the authors restructure the panels, having projective measurements in a brief part and then the proposed method in another part, with no mixing. Also, there are few details in panel b) on how the sparse representation of the data, which is the key idea here, is generated. I suggest to provide more details in this part.

Fig. 2: there are three different setups, but it is not straightforward how they are connected. Can the authors simplify here? Because it is a single-shot imaging system, so I expect a single setup?

Response: We thank the Reviewer for the useful suggestions. We have restructured as far as possible both Figure 1 (change 6) and Figure 2 (change 7) to make them easier to read. Specifically, in Fig. 1 we separated conventional and single-shot polarimetry. We keep a direct visual comparison between the methods to underline the advantages of our scheme for the case of vector beams, also adding within the text more details on the optical transformation of the data. We now show in Fig. 2 a single experimental setup with the vector beam generator connected to the single-shot polarimeter. As the optical line for reference measurements is a well-known apparatus, it has been removed to simplify the setup.

Report #3: 1) I find the reported prediction accuracies very noteworthy results. I suggest that the authors comment on those in the abstract to improve the work's quantitative analysis.

Response: The Reviewer is right. We have now underlined the remarkable accuracy of our method directly in the abstract. (change 8)

Report #3: 2) In the ridge regression problem, how do the authors estimate the condition number c ?

Response: We vary the regularization parameter across several orders of magnitude, and we find no significant differences in testing accuracy. The reported results are for $c=1$.

Report #3: 3) From my understanding, the authors directly train the data's sparse representation by linear regression. I think a possibly exciting way to improve the prediction capability of this system is to use statistical inference on the sparse data generated by the scattering medium. This step could be performed, e.g., with a probabilistic autoencoder. Is this a possibility? Can the authors comment on this point?

Response: The proposed idea is extremely interesting. We use supervised learning but we agree that the most exciting direction is to make the device operate via unsupervised learning, in particular probabilistic generative models such as variational autoencoders. We can think of an architecture that provides statistical distributions for the Stokes parameters from the scattered intensity. This possibility has been highlighted directly in the manuscript conclusion. (change 9)

Reviewer #1 (Remarks to the Author):

Unfortunately, the authors did not address my concerns. Adding a few references here and there and a superfluous sentence in the end is not sufficient to create the factual context for this work.

In the authors' own words, "The idea is to map the beam polarizations into a complex spatial distribution of intensity ...". Well, the "idea" that heavy multiple scattering can be made to act as optical instruments has been around for quite some time. The reference to the paper in PhysicaA 1990 is perhaps too old and can be conveniently ignored along with many other significant works.

The digital machine learning algorithm that the authors use is a possible technical solution to implement this "idea". It is an application of previously developed algorithms of supervised learning to a specific physical problem. It should be presented as such. Using the transfer function, for example, could be another solution for this "idea" of complex mapping due to the massive linear system describing multiple scattering. Discussing these aspects means to present a specific technical development from the appropriate perspective.

And, by the way, it should be well understood that placing the work in the correct circumstances does not diminish the "drastic impact" of "photonic machine learning for a challenging optical application" as claimed.

Reviewer #2 (Remarks to the Author):

I have carefully read the replies of the authors to all reviewers, and I think that they have answered properly to all comments/criticisms raised by the reviewers. Therefore, I can only confirm my first impressions about the manuscript and strongly support its publication in Nature Photonics.

Reviewer #3 (Remarks to the Author):

The authors have provided a satisfactory revision. I recommend publication.

Response to Reviewer #1

Report #1: Unfortunately, the authors did not address my concerns. Adding a few references here and there and a superfluous sentence in the end is not sufficient to create the factual context for this work.

In the authors' own words, "The idea is to map the beam polarizations into a complex spatial distribution of intensity ...". Well, the "idea" that heavy multiple scattering can be made to act as optical instruments has been around for quite some time. The reference to the paper in PhysicaA 1990 is perhaps too old and can be conveniently ignored along with many other significant works.

The digital machine learning algorithm that the authors use is a possible technical solution to implement this "idea". It is an application of previously developed algorithms of supervised learning to a specific physical problem. It should be presented as such. Using the transfer function, for example, could be another solution for this "idea" of complex mapping due to the massive linear system describing multiple scattering. Discussing these aspects means to present a specific technical development from the appropriate perspective.

And, by the way, it should be well understood that placing the work in the correct circumstances does not diminish the "drastic impact" of "photonic machine learning for a challenging optical application" as claimed.

Response: We thank the Reviewer for the second review of our manuscript. Although the Reviewer is not satisfied with the revisions, we suppose they now find clear the novelty and impact of our work since they do not reply with technical arguments to our discussion on the originality of the work.

We understand the Reviewer would have preferred we interpret our results within the context of multiple scattering, but we prefer keeping the introduction focused on the problem of polarization imaging and on photonic machine learning. The reason is that scattering from the disordered medium forms only a part of our scheme, while the method we introduce is much more general than a disorder-based optical instrument. For example, in our machine-learning approach, a different optical process such as the transmission through a layered or a nonlinear material can replace multiple scattering. This said, we have taken into account the Reviewer's suggestion and added a referenced paragraph that focuses on how multiple scattering has been harnessed for various optical instruments and applications. (change 1)

Reviewer #1 (Remarks to the Author):

I have examined the revised manuscript and the authors' rebuttal letter. Unfortunately, the couple of sentences added do not change the fact that mapping a distribution of polarization states into a complex spatial distribution of intensities cannot be claimed to be a new idea. I continue to believe that, even though the use of machine learning to this specific polarimetry application is new, the scientific innovation is rather limited.

Response to Reviewer #1

Report #1: I have examined the revised manuscript and the authors' rebuttal letter. Unfortunately, a couple of sentences added do not change the fact that mapping a distribution of polarization states into a complex spatial distribution of intensities cannot be claimed to be a new idea. I continue to believe that, even though the use of machine learning to this specific polarimetry application is new, the scientific innovation is rather limited.

Response: We thank the Reviewer for further examination of our manuscript. Although we believe that a scientific method can certainly be "claimed to be a new idea" if the same method has never been published before, in the final paper we avoid any explicit claim on the idea under debate. In this way, the reader will be free to assess on their own knowledge the novelty and significance of the concepts presented in the work.